# Nuclear Factor-κB Overexpression is Correlated with Poor Outcomes after Multimodality Bladder-Preserving Therapy in Patients with Muscle-Invasive Bladder Cancer

**DOI:** 10.3390/jcm8111954

**Published:** 2019-11-13

**Authors:** Yun Chiang, Chung-Chieh Wang, Yu-Chieh Tsai, Chao-Yuan Huang, Yeong-Shiau Pu, Chia-Chi Lin, Jason Chia-Hsien Cheng

**Affiliations:** 1Graduate Institute of Oncology, National Taiwan University College of Medicine, Taipei 10002, Taiwan; b93401108@ntu.edu.tw; 2Division of Radiation Oncology, Department of Oncology, National Taiwan University Hospital, Taipei 10002, Taiwan; 3Department of Pathology, National Taiwan University Hospital, Taipei 10002, Taiwan; wangchungchieh@ntuh.gov.tw; 4Division of Medical Oncology, Department of Oncology, National Taiwan University Hospital, Taipei 10002, Taiwan; yctsai@ntuh.gov.tw; 5Department of Urology, National Taiwan University Hospital, Taipei 10002, Taiwan; cyh0909@ntuh.gov.tw (C.-Y.H.); pu5249@ntuh.gov.tw (Y.-S.P.); 6Graduate Institute of Clinical Medicine, National Taiwan University College of Medicine, Taipei 10002, Taiwan

**Keywords:** bladder cancer, bladder preservation, nuclear factor-kappa B, p16, radiotherapy

## Abstract

The aim of this study was to investigate prognostic molecular targets for selecting patients with muscle-invasive bladder cancer undergoing bladder-preserving therapy. Pretreatment biopsy samples from patients with muscle-invasive bladder cancer receiving trimodality bladder-preserving therapy were analyzed for expression levels of p53, p16, human epidermal growth factor receptor-2 (Her-2), epidermal growth factor receptor (EGFR), nuclear factor-kappa B (NFκB; p65), E-cadherin, matrix metalloproteinase-9 (MMP9), meiotic recombination 11 homolog (MRE11), programmed death-1 ligand (PD-L1), and mismatch repair proteins (MLH1, PMS2, MSH2, and MSH6) by immunohistochemical (IHC) staining. The correlations between these molecular markers with local progression-free survival (LPFS), distant metastasis-free survival (DMFS), and overall survival (OS) were explored. Biopsy samples from 41 out of 60 patients were evaluated using IHC. Univariate analysis revealed that the high expression of NFκB is associated with significantly worse LPFS, DMFS, and OS, and low expression of p16 is associated with significantly lower LPFS. Upon further multivariate analysis including sex, age, stage, and selected unfavorable factors in the model, NFκB and p16 independently remained significant. The investigational in vitro study demonstrated that irradiation induces up-regulation of NFκB signaling. Irradiated bladder cancer cells showed increased invasion capability and clonogenic survival; inhibition of NFκB signaling by an NFκB inhibitor, SC75741, or RNA interference reversed the observed increases. NFκB expression (p65) is associated with prognostic significance for both LPFS and DMFS in patients treated with bladder-preserving therapy, with consistent impact on cell viability of bladder cancer cells. NFκB may be a putative molecular target to help with outcome stratification.

## 1. Introduction

Although radical cystectomy is viewed as the standard treatment for muscle-invasive bladder cancer, post-operative urinary diversion may result in major physical and psychological changes with a consequent negative impact on quality of life [1,2,3]. Long-term follow-ups from several institutions have shown that a trimodality treatment consisting of visually complete transurethral resection of bladder tumor (TURBT) followed by chemotherapy and radiotherapy is associated with an excellent chance for long-term survival with the preserved bladder [4,5,6,7,8].

Clinical factors, including T3–T4 disease, incomplete tumor resection, hydronephrosis, and pelvic lymph node involvement, are poor prognosticators of bladder-preserving therapy [5,9,10]. Among these factors, patients with hydronephrosis and/or pelvic lymph node involvement are typically contraindicated for bladder preservation. Our previous study revealed significantly lower distant control, but no worse bladder preservation rate, in patients with unfavorable factors treated with bladder-preserving therapy [11]. Distant failure is also the primary pattern in the patients with the aforementioned poor prognosticators after radical cystectomy [9]. Therefore, clinical prognosticators alone are not sufficient for the selection of candidates with bladder preservation. Given the current prognostic factors or selection criteria for patients with bladder-preserving therapy with exclusively clinical factors (T2–T3, absence of carcinoma in situ, hydronephrosis, or lymphadenopathy), the tumor-associated molecular markers might have prognostic implication for successful organ preservation.

Increasing evidence is showing that various molecular markers, such as p53 [12,13,14,15,16,17,18], p16 [19,20], epidermal growth factor receptor (EGFR) [21,22], human epidermal growth factor receptor-2 (Her-2) [21,22,23], nuclear factor-kappa B (NFκB) [23,24,25], E-cadherin [25], matrix metalloproteinase-9 (MMP9) [26], meiotic recombination 11 homolog (MRE11) [27,28,29], and programmed death-1 ligand (PD-L1) [30], may help predict the outcome of bladder cancer. However, the association between these biomarkers and the outcomes of patients receiving bladder-preserving therapy remains unclear. A recent study [31] showed that mismatch repair (MMR)-deficiency in 12 solid tumor types was more sensitive to immune checkpoint blockade as shown by the development of mutant neopeptides. With radiation as a potential strategy for generating tumor neoantigens [32], MMR status may play a role in patients who receive bladder-preserving trimodality therapy composed of radiotherapy.

The goal of this study was to investigate biomarkers of pre-treatment tumor samples in patients with bladder cancer undergoing trimodality bladder-preserving therapy, and to evaluate the correlation with treatment outcome and patterns of failure. The impacts of selected markers were analyzed using in vitro protein expression and cell viability from two bladder cancer cell lines. T24 is a cell line established from a high-grade and invasive human urinary bladder cancer patient, which consists of one of the major characteristics of muscle-invasive bladder cancer [33]. MB49 is one of the most commonly used murine bladder carcinoma cell lines derived from C57BL/6 mouse [34]. MB49 shares several pivotal tumor characteristics with human bladder cancer, such as cell surface markers, sensitivity to apoptosis, and immunological profile [35].

## 2. Methods and Materials

### 2.1. Patient Sample Collection

Patients with muscle-invasive bladder cancer who underwent maximal TURBT and induction chemotherapy followed by concurrent chemoradiotherapy (CCRT) were retrospectively enrolled from September 2002 to October 2015. Unfavorable factors, including hydronephrosis and/or pelvic nodal involvement, did not preclude patient eligibility. Patients received trimodality bladder-preserving therapy as previously described [11]. Briefly, patients received three cycles of cisplatin-based induction chemotherapy, followed by consolidative CCRT. The radiotherapy doses and fields were 45 Gy to the pelvis, 50.4 Gy to the bladder, and 64.8 Gy to bladder tumor bed with a daily fraction of 1.8 Gy. The bladder tumor tissues were collected prior to induction chemotherapy.

### 2.2. Immunohistochemical (IHC) Staining Analysis

Expressions of p53, p16, EGFR, Her-2, NFκB (p65), E-cadherin, MMP9, MRE11, and PD-L1 were assessed using IHC staining. Pre-treatment with proteinase K (5 min at room temperature) was used for antigen retrieval. Samples were then incubated with primary antibodies (Appendix A). IHC staining was performed using Ventana Benchmark XT (Ventana Medical Systems, Tucson, AZ, USA) according to the manufacturer’s protocol. Slides were counterstained with hematoxylin and permanently mounted.

### 2.3. Interpretation of IHC Staining

For p53, p16, NFκB (p65), and MMR proteins (MLH1, PMS2, MSH2, and MSH6), percentages of positive nuclear immunoreactivity in the tumor were recorded for each sample [36,37]. Positive staining for E-cadherin was defined as the proportion of tumor cells with membranous staining greater than 90%. The results of EGFR and Her-2 staining were semi-quantified as 0, 1, 2, or 3 as the ASCO/CAP 2013 criteria for breast cancer [21,38]. NFκB (p65), p16, p53, EGFR, Her-2, MRE11,28, MMP9, immune cell (IC) of PD-L1, and tumor cell (TC) of PD-L1 [30,39] were scored using defined criteria (Appendix A). Briefly, a score of 0 or 1 was considered negative and 2 or 3 as positive for p16, EGFR, Her-2, and E-cadherin. For p53, a score of 0 or 4 was considered positive and 1–3 as negative. The cut-off point of positivity for NFκB (p65), MMP9, and IC of PD-L1 was 0%. The staining pattern among the cases with positive NFκB staining was scored as 1 (scattered single cells), 2 (patchy with sharp demarcation), 3 (patchy with gradual change), or 4 (diffuse and homogenous). For MMR proteins, loss of staining was designated as complete loss of nuclear staining in all tumor nuclei (Appendix A) [40]. All IHC-stained slides were assessed using light microscopy by the same pathologist who was blinded to the study endpoints to eliminate inter-observer variability.

### 2.4. Bladder Tumor Tissue Collection

Formalin-fixed and paraffin-embedded blocks of bladder and tumor tissues from TURBT before induction chemotherapy were collected. We took 5 µm sections from each tissue block for IHC staining.

### 2.5. Cell Culture

Human bladder urothelial carcinoma cell line, T24, and murine bladder tumor cell line, MB49, were purchased from the ATCC/Bioresource Collection and Research Center (BCRC). T24 cells were cultured in DMEM supplemented with 10% FBS. MB49 cells were cultured in RPMI-1640 medium with 10% FBS. Cells were incubated at 37 °C in a humidified atmosphere (5% CO_2_ and 95% air). The MB49 knockdown cell line targeting NFκB subunit p50 protein (NFκB KD) (Appendix A) with RNA interference (RNAi) was purchased from Academia Sinica (Clone ID: TRCN0000009511; Taipei, Taiwan).

### 2.6. Reagents

SC75741, an NFκB inhibitor involving impaired DNA binding of the NFκB subunit p65 [41], was purchased from Selleck Chemicals (Houston, TX, USA). A Western blot assay was conducted with bladder cancer cells exposed to ionizing irradiation of 5 Gy either alone or combined with 5 μM of SC75741 for 24 h. Cells were treated with different doses of SC75741 and irradiation, and incubated for 7 days for colony formation assays.

### 2.7. Western Blot Assay

Aliquots of cell lysates containing 50 μg of protein were separated using SDS-PAGE (6%–15% polyacrylamide), transferred onto polyvinylidene difluoride membranes and immunoblotted with primary antibodies directed against p65, phospho-p65 (p-p65), and p50 (Cell Signaling Technology, Danvers, MA, USA). Bound antibodies were detected using appropriate peroxidase-coupled secondary antibodies followed by enhanced chemiluminescence.

### 2.8. Immunofluorescence Staining

MB49 cells and MB49 NFκB KD cells were plated on polylysine-coated coverslips, allowed to attach overnight, and exposed to ionizing irradiation (5 Gy). After irradiation, cells were incubated for 0, 1, or 3 h. Cells were washed three times with cold phosphate-buffered saline (PBS), followed by fixation with 4% formaldehyde/PBS for 30 min, permeabilized in 0.5% Triton X-100 in PBS for 30 min, and blocked in 5% bovine serum albumin (BSA) for 30 min at room temperature. Fixed cells were incubated overnight at 4 °C with anti-phospho-p65 (Cell Signaling Technology, Danvers, MA, USA). Cells were washed and incubated with FITC-conjugated anti-rabbit immunoglobin G (IgG) antibody (Bethyl Laboratories, Inc., Montgomery, TX, USA) for 90 min at room temperature in the dark. Nuclei were stained with DAPI for 5 min. Cells were examined using a fluorescent microscope. The experiment was repeated four times.

### 2.9. Colony Formation Assay

Cells (500/well) were seeded in six-well plates and treated with several doses of irradiation (2.5–10 Gy) following 24 h pretreatment with various doses of SC75741 (1–5 μM). Cells were then cultured for an additional 7 days, after which the number of colonies (clusters of more than 50 cells) visible to the naked eye were counted in each well. At each drug concentration, the surviving fraction was determined by dividing the total number of colonies after irradiation by the number of colonies without irradiation.

### 2.10. Cell Invasion Assay

The capacity for cell invasion was evaluated using a Boyden Chamber assay. Chambers with an 8 μm diameter pore were obtained from Corning Inc. (Corning, NY, USA). The chambers were placed in 24-well plates with 500 μL RPMI per well, and a 100 μL cell suspension with 100,000 cells was added in the upper chamber containing 50 μL mixture of FBS-free RPMI/Matrigel at 8/1 ratio (BD Bioscience, San Jose, CA, USA). After a 4 h incubation at 37 °C, chambers were either irradiated or not. Then, 48 h following irradiation, the chambers were washed with PBS (pH 7.4) three times to remove the cells in the upper chamber and stained with crystal violet (0.5% in ethanol) for 20 min followed by PBS (3×). The cells were counted using a microscope, five fields were randomly chosen at 200 × magnification, and the average number of cells was analyzed

## 3. Results

### 3.1. Patient Characteristics

Available IHC data were collected from 41 (IHC group) out of 61 patients from September 2002 to October 2015. Nine patients received TURBT at outside hospitals before referral, and 11 patients had an insufficient amount of tissue for this study. Baseline characteristics were similar between the IHC group and no IHC group (Table 1 and Table 2). All patients had high-grade urothelial carcinoma and achieved complete response on cystoscopy after induction chemotherapy. During the follow-up period (median: 87 months, range: 51–124 months), 10 (24%) patients died, 11 (27%) patients experienced local recurrence, six (15%) patients had distant metastasis, and two (5%) patients had both local recurrence and distant metastasis. The rates of the defined immunoreactivity for the markers are listed in Table 2 (Appendix A).

### 3.2. Prognostic Significance of Molecular Biomarker

Among the examined molecular markers, univariate analysis revealed that positive immunoreactivity of nuclear, but not cellular NFκB, was significantly associated with shorter LPFS (*p* = 0.05), DMFS (*p* = 0.03), and OS (*p* = 0.04) (Appendix A, Figure 1A–C). Further stratification of staining pattern among cases with positive nuclear NFκB staining showed no significant difference in clinical outcome. Negative staining of p16 was significantly correlated with lower LPFS (*p* = 0.01) (Appendix A, Figure 1D), but not DMFS (*p* = 0.1) or OS (*p* = 0.1) (Appendix A). The trend in negative staining of p53 with lower LPFS, DMFS, and OS was noted (Appendix A), but subsequent integration of p53 into a panel of biomarkers (p53/p16, p53/NFκB, p53/p16/NFκB) was not associated with clinical outcomes (data not shown). After adjusting for other clinical prognostic factors including sex, age, T stage, and unfavorable group, multivariate analysis showed that both NFκB (*p* = 0.001), p16 (*p* = 0.002), and p53 (*p* = 0.005) are independent factors for LPFS, and only NFκB remained significant for DMFS (*p* = 0.01) and OS (*p* = 0.014) (Appendix A).

### 3.3. Comparisons of Immunostaining of NFκB between Pre-Treatment Samples and Recurrent Samples

Eight out of 20 patients who experienced local recurrence and/or distant metastasis had available TURBT specimens of both initially diagnosed and recurrent bladder tumors. Six patients were negative for p16 and positive for NFκB at diagnosis. IHC staining of p16 and NFκB were further assessed in recurrent bladder tumors. Five patients were persistently negative for p16, but had increased immunoreactivity of NFκB in their recurrent tissues compared with the pre-treatment tissues (Figure 2 and Appendix A). One patient had similar NFκB and p16 staining. Despite the limited number of recurrent tissue samples, these findings suggest that overexpression of NFκB might be associated with bladder tumor recurrence.

### 3.4. In Vitro Investigation of NFκB Expression of Irradiated Bladder Cancer Cells

Given the findings by IHC staining of disease recurrence associated with the up-regulation of NFκB in patients undergoing trimodality treatment for bladder cancer, we next investigated the correlation between irradiation and NFκB signaling pathway with human (T24) and murine (MB49) bladder cancer cell lines. Radiation-activated NFκB signaling presented as increased p-p65, one of the major subunits of NFκB. The expression was reduced by an NFκB inhibitor, SC75741 (Figure 3). In terms of cell viability, a colony formation assay showed a dose-dependent decrease in clonogenic survival in irradiated bladder cancer cells with NFκB signaling repressed by SC75741 (Figure 3). Inhibition of NFκB signaling using RNAi in the MB49 NFκB KD cell line, confirmed by interference of nuclear translocation of p-p65 shown on immunofluorescence, resulted in decreased invasion capability evaluated using the Boyden chamber assay (Figure 4). These results indicate that inhibition of radiation-induced activation of NFκB may have a negative impact on the invasiveness and proliferation of bladder cancer cells. 

With radiation-induced NFκB activation, we conducted additional experiments to identify the subsequent impact of NFκB activation on the tumor microenvironment for the underlying mechanism. We found a decreased interleukin (IL)-1β concentration in irradiated wild-type MB49 bladder cancer cells of the culture medium. If NFκB signaling was inhibited by RNA interference using MB49 NFκB knockdown cells, increased IL-1β concentration was shown in culture medium (Appendix A). With the recombinant IL-1β (Invitrogen, Waltham, MS, USA) added into the culture medium of MB49 cells for determining the potential impact on survival of bladder cancer cells, clonogenic survival in irradiated bladder cancer cells was reduced with the added IL-1β protein in a dose-dependent manner (Appendix A). The IL-1β protein had no significant impact on clonogenic survival of non-irradiated bladder cancer cells (Appendix A). Therefore, we hypothesize that radiation-activated NFκB signaling stimulates tumor progression by modulating post-irradiation cytokine expression, such as IL-1β.

## 4. Discussion

The current study revealed that NFκB overexpression is a negative prognosticator of LPFS, DMFS, and OS, whereas expression of p16 is inversely associated with LPFS in patients with bladder cancer undergoing trimodality bladder-preserving therapy. Further in vitro investigation demonstrated that ionizing irradiation induced activation of NFκB signaling and promoted cell survival and invasiveness of bladder cancer cells. A previous study on putative IHC biomarkers in pre-treatment biopsy samples of patients with muscle-invasive bladder cancer, who received induction chemoradiation followed by cystectomy, showed that overexpression of NFκB is associated with chemoradiation resistance and lower cancer-specific survival, in line with our result [23].

Most prognostic indicators of bladder preservation that have been investigated are clinical factors, such as hydronephrosis, pelvic nodal involvement, T stage, and completeness of TURBT [5,9,10]. However, we previously found that hydronephrosis and pelvic nodal involvement, defined as combined unfavorable factors, are associated with distant metastasis but not local recurrence [11]. This finding may indicate a reduced role for clinical factors in selecting candidates for successful bladder preservation. Although many molecular markers have been reported for predicting outcomes of patients with advanced bladder cancer [12,13,14,15,16,17,18,19,20,21,22,23,24,25,26,27,28,29,30], fewer studies [12,13,29] focused on the biomarkers in the specific subgroup of patients with bladder preservation strategies. The current study reveals that clinical factors are not as significant in their association with survival outcomes as molecular biomarkers in patients undergoing bladder preservation.

NFκB, a heterodimeric complex with two major subunits of p65 and p50 proteins, is a transcription factor associated with anti-apoptosis, angiogenesis, proliferation, and distant metastasis in tumor cells [42,43,44]. The regulatory function of NFκB involves phosphorylation of its subunits, such as p65, for promoting nuclear translocation of NFκB and binding to enhancer/promoter regions of target genes [45]. NFκB overexpression has been identified as a risk factor of chemoradiation resistance in muscle-invasive bladder cancer [23]. Previous in vitro studies revealed increased migration, angiogenesis, and metastasis of human bladder cancer cells mediated by NFκB signaling [46,47]. Karashima et al. [46] showed that in vivo suppression of NFκB through injection of mutant IκB human bladder cancer cells into the bladder wall of nude mice inhibited bladder tumor growth and lymph node metastasis. In the current study, NFκB overexpression is associated with decreased LPFS, DMFS, and OS in patients undergoing bladder preserving therapy consisting of TURBT, chemotherapy, and radiotherapy. This is consistent with its role in both local and distant progression. The increased immunoreactivity of NFκB in recurrent tumor samples compared with that in pre-treatment samples suggests that NFκB activation is correlated with bladder cancer recurrence. With radiation-induced NFκB activation restrained by SC75741 or RNAi, decreased clonogenic survival and capacity for invasion of bladder cancer cells were shown. The translational data further support the clinical correlation of NFκB expression and tumor progression in patients with bladder cancer. Furthermore, additional experiments suggest radiation-activated NFκB signaling might stimulate bladder tumor progression by regulating the expression of IL-1β.

Ionizing radiation induces NFκB signaling and DNA binding activity with various intensities in different tissues [48,49]. Sustained NFκB activation could facilitate cells with accumulated radiation-induced damage to escape elimination by apoptosis and contribute to radio-resistance [44]. An in vitro study demonstrated that constitutive NFκB activity prevented prostate cancer cells from apoptosis and resulted in a more aggressive potential for metastasis [50]. A correlation between NFκB overexpression in human IHC-stained bladder tumors and chemo-radio-resistance has been reported [23]. We deduce from this research and our study that radiation may induce activated NFκB signaling, which prevents the subsequent lethal cascade in bladder cancer.

The product of p16, a cyclin-dependent kinase inhibitor, is a well-known tumor suppressor. Down-regulation of p16 has been reported to be associated with increased tumor progression and decreased survival in patients with bladder cancer [19,20,51]. The primary mechanism behind p16 inactivation-related bladder cancer progression is the uncontrolled cell proliferation from cell cycle dysregulation [19,51]. However, no available in vitro studies link p16 to angiogenesis or metastatic potential of bladder cancer cells. This may be one of the reasons for the negative association of p16 expression with only LPFS, but not DMFS. A study showed that increased expression of nuclear p16 is induced by cisplatin and leads to ubiquitination of NFκB in head and neck cancer [52]. Whether p16 inactivation plays a role in overexpression of NFκB remains unsettled. Further in vitro analysis, such as using p16 knock-down cell lines, to investigate the impact of p16 activation on NFκB signaling pathways to disclose the interaction is demanded

Most patients with bladder cancer undergoing bladder preservation therapy have localized bladder tumor(s) eligible for TURBT. Local recurrence may be adequately salvaged (nine of 11 patients with local recurrence were salvaged in the current study), whereas other patterns of failure (e.g., distant metastasis) frequently failed to respond to treatment and translate to a prognostic impact on survival. This clinical situation may partly explain some molecular biomarkers, such as p16, having a significant association with LPFS but not DMFS or OS. The impact of NFκB on distant control emphasizes its importance for both predictive and prognostic roles in patient selection and follow-up with bladder preservation therapy.

Immune checkpoint inhibitors contribute to significantly prolonged survival in patients with metastatic urothelial carcinoma [53,54,55,56], and some researchers have proposed higher expression of PD-L1 as a positive prognosticator [30]. In contrast, PD-L1 protein expression in either IC or TC was not associated with survival outcome in the current study. One possible explanation is that the low tumor volume of patients in the current study in comparison with metastatic disease load might have an uncertain impact on the prognostic value of PD-L1 expression based on its relationship with cancer cell immune evasion [55]. Several technical and analytical limitations of IHC, such as PD-L1 antibody clones, quantitative scores of PD-L1, and the definition of PD-L1 positivity, might also contribute to this controversy.

The current study has some limitations worth noting. The small sample size with limited availability of biopsy specimens might influence the statistical significance and mask the effects of other biomarkers on survival outcomes. The small amount of tissue from TURBT may have had an uncertain impact on the interpretation of staining scores. We used only loss of function in vitro analysis, with NFκB inhibitor or RNAi, which was not sufficient to depict the comprehensive role of NFκB activation in bladder cancer cells. Using a gain of function experiment with NFκB, such as p65-expressing vector, could help delineate the cancer-promoting function of NFκB signaling. The bladder cancer cell lines used in our study, T24 and MB49, are well-established and have been widely used in many in vitro studies investigating the characteristics of bladder cancer might not completely reflect the post-irradiation tumor microenvironment in patients with muscle-invasive bladder cancer. The patient-derived xenograft using tissue from specific patient’s muscle-invasive bladder tumor implanted into an immunodeficient or humanized mouse would be a future goal and an excellent model in a prospective setting. Finally, as growing evidence suggests the interactive roles of NFκB signaling and inflammation in tumorigenesis [57], more IHC tests of biomarkers associated with immune cells, such as tumor-associated macrophage, and an in vivo investigation would help clarify the association of radiation on the tumor microenvironment and NFκB activation. This study started with the discovery of molecular markers from tissue IHC for their association with survival outcomes of patients with muscle-invasive bladder cancer undergoing uniform bladder-preservation therapy. The translational part following the IHC findings with NFκB confirmed that radiation-activated NFκB signaling is associated with resistance of bladder cancer cells. The underlying mechanism on additional radiation-activated NFκB to the pre-existing overexpressed NFκB signaling remains to be explored. Notably, a combinatory impact of multiple events, rather than a single consequence of NFκB signaling pathway, results in bladder tumor progression. Besides the interactive roles of NFκB signaling and tumor microenvironmental inflammation in tumorigenesis, more investigational work is ongoing for the downstream consequences/targets of NFκB. 

A particularly intriguing NFκB target gene encodes pro-IL-1β, which is processed by caspase-1 or neutrophil protease to the key proinflammatory and tumor-promoting cytokine IL-1β. To promote pro-IL-1β expression, NFκB also negatively controls its processing to mature IL-1β through the induction of various protease inhibitors and contributes to the self-limiting nature of inflammatory signaling [57]. Notably, dual roles of IL-1β have been reported. Sustained IL-1β may promote tumor propagation by different mechanisms [58]. On the other hand, IL-1β production might be associated with anti-tumor capacity in some circumstances [59], which is consistent with our finding.

With tissues of recurrent tumors available from only eight patients, we found five of them with increased expression of NFκB. We understand the limitation in proposing the association due to the small number of patients with available recurrent tumor tissue, but would like to demonstrate the post-treatment increased NFκB of these recurrent tissues for the potential implication. Pre-treatment but not post-treatment IHC staining for NFκB was the significant factor associated with the survival outcomes in our data analysis. Therefore, we did not expand the interpretation from the limited number of patients with available recurrent tumor tissues. To propose the potential link, we supplemented the in vitro data on irradiated bladder cancer cells with data about the increased NFκB expression and enhanced invasiveness and clonogenic survival. These findings might still imply the importance of radiation-activated NFκB signaling for the resistance of bladder cancer, and support the rationale for future intervention of NFκB for bladder preservation with radiotherapy and other treatments.

In conclusion, the current results demonstrate the prognostic value of NFκB and p16 in patients with bladder cancer who received bladder-preserving therapy. Increased immunoreactivity of NFκB, as confirmed by IHC staining, indicated more lethal cancer features including local recurrence and distant metastasis, which demand more aggressive treatment for this subgroup of patients. The strategy to inhibit NFκB deserves further investigation and incorporation into a therapeutic approach for bladder-preserving protocols.

## Figures and Tables

**Figure 1 jcm-08-01954-f001:**
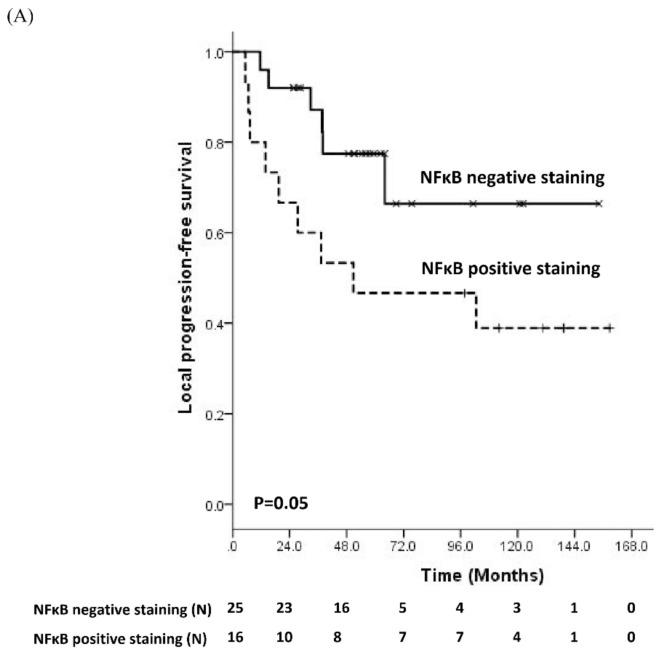
Positive NFκB staining is associated with lower rates of local progression-free survival (LPFS), distant metastasis-free survival (DMFS), and overall survival (OS) in patients treated with trimodality bladder-preserving therapy, negative p16 staining was a poorer prognosticator of LPFS. Kaplan–Meier curves of (**A**) LPFS, (**B**) DMFS, and (**C**) OS stratified by the immunohistochemical staining of NFκB. (**D**) Kaplan–Meier curve of LPFS stratified by the immunohistochemical staining of p16.

**Figure 2 jcm-08-01954-f002:**
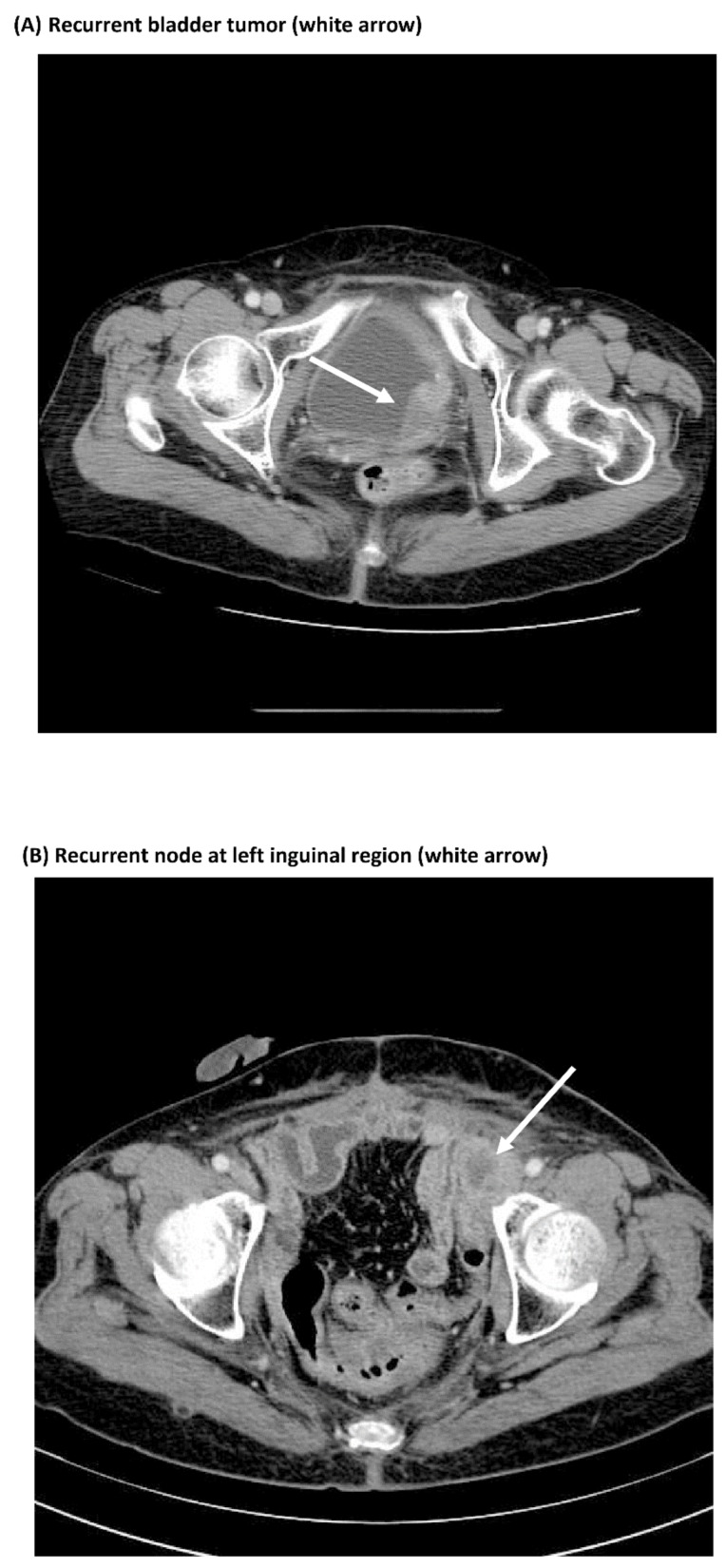
Increased nuclear staining of NFκB in recurrent bladder tumor tissue compared with pre-treatment tumor tissue is shown in a patient with T3N0M0 bladder cancer treated with trimodality bladder-preserving therapy and developing (**A**) local bladder recurrence, (**B**) inguinal node metastasis, and (**C**) mediastinal node metastasis on computed tomography at 7 months, 22 months, and 29 months, respectively. (**D**) Immunohistochemical staining of NFκB and p16 in pre-treatment and recurrent bladder tumors at magnifications of 200× (scale bar: 100 μm).

**Figure 3 jcm-08-01954-f003:**
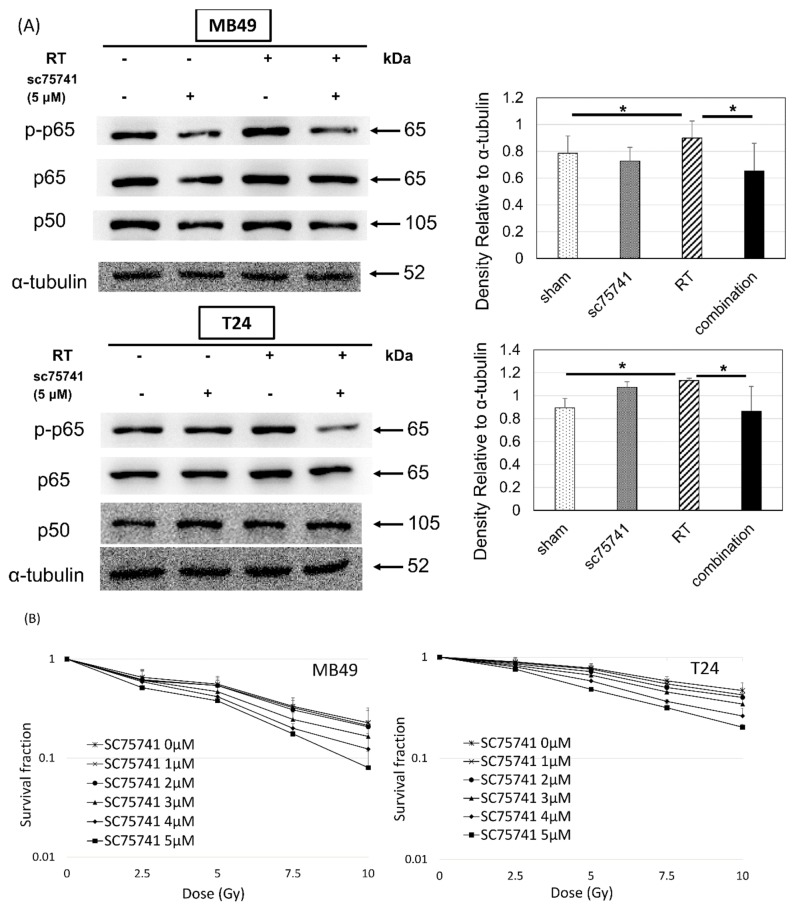
NFκB signaling was up-regulated after irradiation (RT) and inhibited with the treatment of an NFκB inhibitor, SC75741, mitigating the proliferative capability of bladder cancer cells. (**A**) Images and quantification by densitometry and ImageJ of Western blots for NFκB transcription factor proteins (p65 and p50) (n = 3) (n = 3) and (**B**) colony formation assays of MB49 and T24 cancer cells after pre-treatment with or without SC75741 (5 μM) and/or irradiation (5 Gy) (n = 4). * *p* < 0.05.

**Figure 4 jcm-08-01954-f004:**
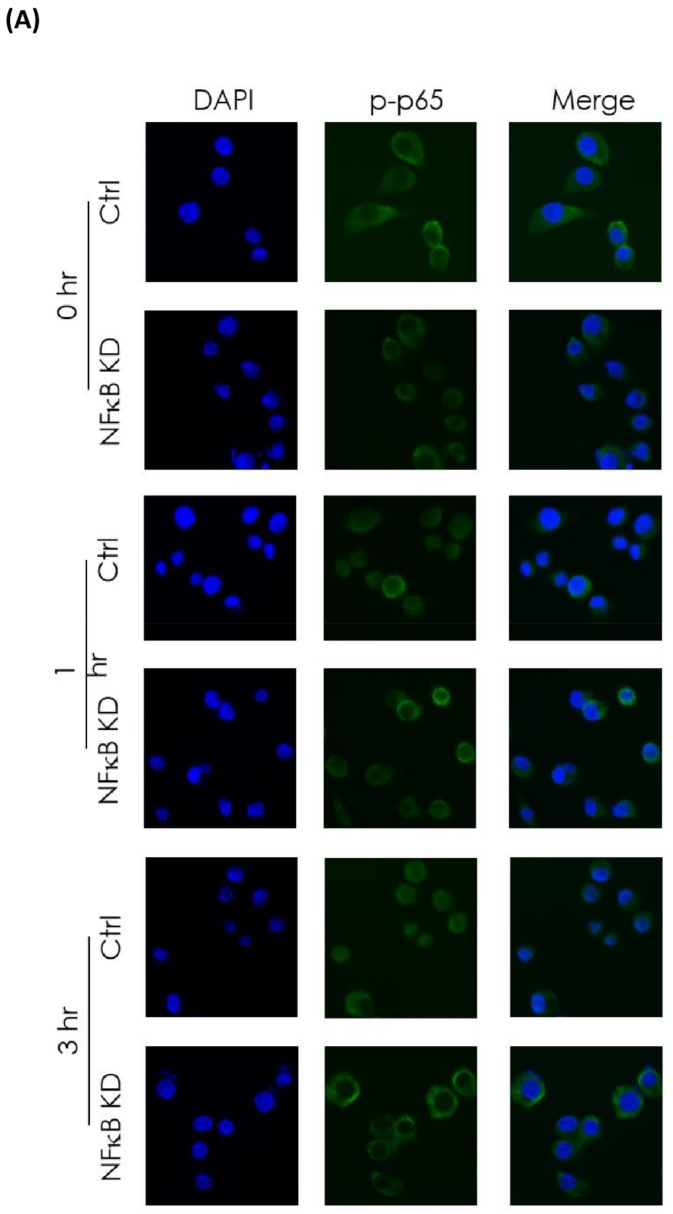
Blockade of NFκB signaling by RNA interference inhibited radiation (RT)-induced nuclear translocation of phosphor-p65 and the invasiveness capability of murine bladder cancer cells. (**A**) Immunofluorescence staining of phospho-p65 at 0, 1, and 3 h after RT (5 Gy) of MB49 vector-control (Ctrl) and MB49 NFκB knock-down (KD) cells. (**B**) MB49 control (Ctrl) and MB49 NFκB KD cells were seeded in Matrigel-coated inserts of Boyden chambers, treated without and with RT (5 Gy). After 24 h, the invading cells were viewed microscopically (high-power field, 200 ×, left panel). Invading cells were counted. * *p* < 0.05, Ctrl sham group vs. Ctrl RT group.

**Table 1 jcm-08-01954-t001:** Comparison of characteristics between patients with immunohistochemical staining data (IHC group, *n* = 41) and non-IHC group (*n* = 20). IHC: immunohistochemistry; CF: cisplatin plus 5-fluorouracil; PCF: paclitaxel plus CF; GC: gemcitabine plus cisplatin; Unfavorable group: hydronephrosis/hydroureter and/or pelvic nodal involvement; Favorable group: other than unfavorable group.

Characteristics		No of Patients (%)		Fisher’s Exact Test or χ^2^ Test *p*-Value
		Non-IHC Group (*n* = 20)	IHC Group (*n* = 41)	
**Sex**	Male	14 (70)	25 (61)	0.35
	Female	6 (30)	16 (39)	
**Age (years)**	<70	16 (80)	24 (59)	0.084
	≥70	4 (20)	17 (41)	
**Clinical T stage**	T2	19 (95)	30 (73)	0.083
	T3–T4a	1 (5)	11 (27)	
**Induction chemotherapy**	CF	4 (20)	12 (29)	0.78
	PCF	5 (25)	11 (27)	
	GC	11 (55)	18 (44)	
**Prognostic factors**	Favorable group	16 (80)	33 (80)	0.61
	Unfavorable group	4 (20)	8 (20)	

**Table 2 jcm-08-01954-t002:** Patient characteristics (*n* = 41). CF: cisplatin plus 5-fluorouracil; PCF: paclitaxel plus CF; GC: gemcitabine plus cisplatin; Unfavorable group: hydronephrosis/hydroureter and/or pelvic nodal involvement); Favorable group: other than unfavorable group; NFκB: nuclear factor-kappa B; EGFR: epidermal growth factor receptor; MMP9: matrix metalloproteinase-9 (MMP9); MRE11: meiotic recombination 11 homolog; PD-L1: programmed death-1 ligand; IC: immune cell; TC: tumor cell; MMR: mismatch repair.

Characteristics		No. of Patients (%)
**Sex**	Male	25 (61)
	Female	16 (39)
**Age (years)**	<70	24 (59)
	≥70	17 (41)
**Clinical T stage**	T2	30 (73)
	T3–T4a	11 (27)
**Induction chemotherapy**	CF	12 (29)
	PCF	11 (27)
	GC	18 (44)
**Risk factors**	Favorable group	33 (80)
	Unfavorable group	8 (20)
**NFκB (p65)**	Negative (≤0)	25 (61)
	Positive (>0)	16 (39)
**p16**	Negative (score 0–1)	24 (59)
	Positive (score 2–3)	17 (41)
**p53**	Negative (score 0 or 4)	25 (61)
	Positive (score 1–3)	16 (39)
**EGFR**	Negative (score 0–1)	26 (63)
	Positive (score 2–3)	15 (37)
**Her-2**	Negative (score 0–1)	29 (71)
	Positive (score 2–3)	12 (29)
**E-cadherin**	Negative (score 0–1)	6 (15)
	Positive (score 2–3)	35 (85)
**MMP9**	Negative (≤0)	30 (73)
	Positive (>0)	11 (27)
**MRE11**	Low (≤median)	21 (51)
	High (>median)	20 (49)
**IC of PD-L1**	Negative (≤0)	24 (59)
	Positive (>0)	17 (41)
**TC of PD-L1**	Negative (≤5%)	29 (71)
	Positive (>5%)	12 (29)
**MMR proteins (PMS2/MLH1/MSH2/MSH6)**	Negative	0
	Positive	41 (100)

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
