# Peer review of "Nuclear Factor-κB Overexpression is Correlated with Poor Outcomes after Multimodality Bladder-Preserving Therapy in Patients with Muscle-Invasive Bladder Cancer"

_jcm, 2019, doi:10.3390/jcm8111954_

Round 1
Reviewer 1 Report
1. The authors improved their findings by increasing the number of patients with recurrent tumors, where they showed increased immunoreactivity of NFκB in two out of four patients. The authors also refined the respective statement regarding a descriptive IHC finding rather than the proposal of a statistical association between NFκB-overexpression and poor outcome after multimodality bladder‐preserving therapy. With the respect to these changes and the refined title, the initial points raised are now properly addressed.
2. The authors now provide a suitable discussion showing the feasibility of their cellular model, but should include the detailed discussion of the response as well as the respective reference also in the discussion or introduction section of their manuscript to better guide the reader.
3. The authors now determined an underlying molecular mechanism in more detail and hypothesize from new experiments that radiation‐activated NFκB signaling stimulates tumor progression by modulating IL‐1β expression. This matter is highly interesting and properly addresses the initial comment. Since IL-1ß is also discussed as a target gene of NF-kB, the authors may include a small section in their discussion regarding the regulation of IL‐1β by NF-kB in cancer to discuss their interesting findings in the context of current literature.
Author Response
We deeply appreciate your kind advice and informative suggestion. In response to the comments from the reviewer, we have made some revisions and resubmit a revised version according to the editorial decision. The comments and our responses are listed as follows
Reviewer #1
The authors improved their findings by increasing the number of patients with recurrent tumors, where they showed increased immunoreactivity of NFκB in two out of four patients. The authors also refined the respective statement regarding a descriptive IHC finding rather than the proposal of a statistical association between NFκB-overexpression and poor outcome after multimodality bladder-preserving therapy. With the respect to theses changes and the refined title, the initial points raised are now properly addressed.
Response:
We thank the reviewer for the previous practical suggestions to improve our research.
The authors now provide a suitable discussion showing the feasibility of their cellular model, but should include the detailed discussion of the response as the respective as well as the respective reference also in the discussion or introduction section of their manuscript to better guide the reader.
Response:
We thank the reviewer for the suggestion. We add the following sentences into the Introduction section (Page 2, Lines 78-82). “T24 is a cell line established from a high-grade and invasive human urinary bladder cancer patient, which consists of one of the major characteristics of muscle-invasive bladder cancer [33]. MB49 is one of the most commonly used murine bladder carcinoma cell lines derived from a C57BL/6 mouse [34]. MB49 shares several pivotal tumor characteristics with human bladder cancer, such as cell surface markers, sensitivity to apoptosis, and immunological profile [35].”
The authors now determined an underlying molecular mechanism in more detail and hypothesize from new experiments that radiation‐activated NFκB signaling stimulates tumor progression by modulating IL-1ß expression. This matter is highly interesting and properly addresses the initial comment. Since IL-1ß is also discussed as a target gene of NF-kB, the authors may include a small section in their discussion regarding the regulation of IL-1ß by NF-kB in cancer to discuss their interesting findings in the context of current literature.
Response:
We appreciate the reviewer’s precious comment. We add the following sentences into the 3rd paragraph of the Discussion section (Page 14, Lines 328-329). “Furthermore, additional experiments suggest radiation-activated NFκB signaling might stimulate bladder tumor progression by regulating the expression of IL-1β.”
We also include a paragraph regarding the IL-1β in the Discussion section (Page 15, Lines 391-397), as the reviewer kindly suggested. “A particularly intriguing NFκB target gene encodes pro-IL-1β, which is processed by caspase-1 or neutrophil protease to the key proinflammatory and tumor-promoting cytokine IL-1β. To promote pro-IL-1β expression, NFκB also negatively controls its processing to mature IL-1β through the induction of various protease inhibitors and contributes to the self-limiting nature of inflammatory signaling [57]. Notably, dual roles of IL-1β have been reported. Sustained IL-1β may promote tumor propagation by different mechanisms [58]. On the other hand, IL-1β production might be associated with anti-tumor capacity in some circumstances [59], which is consistent with our finding.”
Reviewer 2 Report
This manuscript by Cheng et al. reports on the pre-treatment expression of a number of putative molecular prognostic markers for patients with MIBC who underwent trimodal bladder preserving therapy with univariate and multivariate analyses of association of marker expression with LPFS, SMFS and OS.
The key finding from this work is the association of increased NFkB expression with reduced LPFS, DMFS and OS and decreased p16 expression with reduced LPFS. The study is inevitably a little limited by the modest sample size as the authors acknowledge, however, there are statistically significant associations and there is merit in reporting these current findings to the scientific community.
However, there are a number of points that should be addressed before this work would be suitable for publication:-
1) The quality of data presentation needs improving in certain figures before it would be suitable for publication. Specifically:-
a) Font sizes for Figure 1 are extremely small - axes and line labellings, p values.
b) Table 3 - consider presenting data differently to communicate clearly to the reader the most important findings - and place table in the supplementary section
c) Figure 2 D - immunohistochemistry images are too small to be able to claim increased nuclear NFkB staining.
d) Fig. 3A - as well as showing the raw immunoblot images include densitometric analyses and indicate number of biological repeats. RT is non-standard abbreviation and not defined in the figure legend.
e) Fig 3B font sizes /fig. quality is poor.
f) Fig. 4 - immunofluorescent images are too small, merged images do not show presence of p-p65 in the nucleus
g) Fig 4B - images are too small, of low resolution and difficult
2) DMFS and OS Kaplan-Meier curves should also been shown for p16.
3) NFkB is a protein complex consisting of p65 and p50 - what are you actually detecting in your IHC analyses - presumably p65 - this should be made clear. Cellular localisation of NFkB is also clearly very important - what are you positively scoring - increased cellular staining? increased nuclear staining? are there differences in localisation between positive scoring samples?
4) line 350 - the authors suggest that overexpression of NFkB might be due to p16 inactivation - can the authors present any data to support this? eg. serial sections stained for p65 and p16
5) How does the NFkB inhibitor work? This should be made clear in the text - is this supported by the experimental data?
6) Fig 4 - important control is missing - show that p50? is reduced in the KD cells both by immunoblotting and by IF - why are these referred to as NFkB KD cells? What are the consequences of p50 KD on NFKB signalling? p65 total levels, p-p65 and localisation and transactivation of NFkB gene targets?
7) NFkB signalling has many downstream consequences/targets that could be associated with worse prognosisis and reported in vitro observations. This reviewer is confused by the focus on IL1B and the TME and claims made in lines 280-282. Other downstream targets of NFKB should also be analysed.
8) minor typological errors/proof reading required eg. cells were not cultured in 95% oxygen..!
Author Response
This manuscript by Cheng et al. reports on the pre-treatment expression of a
number of putative molecular prognostic markers for patients with MIBC who
underwent trimodal bladder preserving therapy with univariate and multivariate
analyses of association of marker expression with LPFS, SMFS and OS.
The key finding from this work is the association of increased
NFkB expression with reduced LPFS, DMFS and OS and decreased p16
expression with reduced LPFS. The study is inevitably a little limited by the
modest sample size as the authors acknowledge, however,
there are statistically significant associations and there is merit in reporting
these current findings to the scientific community.
However, there are a number of points that should be addressed before this
work would be suitable for publication:-
1) The quality of data presentation needs improving in certain figures before it
would be suitable for publication. Specifically:-
a) Font sizes for Figure 1 are extremely small - axes and line labellings, p
values.
Response:
We appreciate the reviewer’s suggestion. We revise the Figure 1 as below.
b) Table 3 - consider presenting data differently to communicate clearly to the
reader the most important findings - and place table in the supplementary
section
Response:
We appreciate the reviewer’s suggestion. We move the table 3 to the
supplementary section and re-named it as Table S3
c) Figure 2 D - immunohistochemistry images are too small to be able to claim
increased nuclear NFkB staining.
Response:
We appreciate the reviewer’s suggestion. We revise the Figure 2D as below.
d) Fig. 3A - as well as showing the raw immunoblot images include
densitometric analyses and indicate number of biological repeats. RT is nonstandard abbreviation and not defined in the figure legend.
Response:
We appreciate the reviewer’s suggestion. We provide the raw images of
Western blots and revise the Figure 3A with additional densitometric analyses
as below. We also modify the figure legend with the indicated number of
biological repeats and full name of irradiation as below: “NFκB signaling was
upregulated after irradiation (RT) and inhibited with the treatment of an
NFκB inhibitor, SC75741, mitigating the proliferative capability of bladder
cancer cells. (A) Images and quantification by densitometry and ImageJ
of Western blots for NFκB transcription factor proteins (p65 and p50) (n=3)
and (B) colony formation assays of MB49 and T24 cancer cells after pretreatment with or without SC75741 (5 μM) and/or irradiation (5 Gy) (n=4).
*P< 0.05.”
(A) The raw images of Western blot
65
65
105
p-p65
p65
p50
MB49 T24
(B) Revised Figure 3A.
e) Fig 3B font sizes /fig. quality is poor.
Response:
We appreciate the reviewer’s suggestion. We revise the Figure 3B as below.
f) Fig. 4 - immunofluorescent images are too small, merged images do not
show presence of p-p65 in the nucleus
Response:
We appreciate the reviewer’s suggestion. We revise the Figure 4A as below.
g) Fig 4B - images are too small, of low resolution and difficult
Response:
We appreciate the reviewer’s suggestion. We revise the Figure 4B as below.
2) DMFS and OS Kaplan-Meier curves should also been shown for p16.
Response:
We appreciate the reviewer’s suggestion. We add the DMFS and OS survival
curves for p16 as below and in the Result (Page 6, Line 214) and
Supplementary section (Figure S3).
3) NFkB is a protein complex consisting of p65 and p50 - what are you actually
detecting in your IHC analyses - presumably p65 - this should be made clear.
Cellular localisation of NFkB is also clearly very important - what are you
positively scoring - increased cellular staining? increased nuclear staining? are
there differences in localisation between positive scoring samples?
Response:
We appreciate the reviewer’s suggestion. We detected p65 as the NFκB
expression in our IHC analyses. We revise the NFκB as NFκB (p65) in the
material section (Page 3, Line 95; Page 3, Line 103-111), table (Table 1), and
supplementary table (Table S1-S3) for specifying the staining pattern. We
analyzed both the nuclear staining and cellular staining of p65, but only
increased nuclear staining of p65 was associated with survival outcome. We
added this information in the Results section (Page 6, Lines 209-213)
“Among the examined molecular markers, univariate analysis revealed
that positive immunoreactivity of nuclear NFκB, but not cellular NFκB,
was significantly associated with shorter LPFS (p = 0.05), DMFS (p = 0.03),
and OS (p = 0.04) (Table S3, Figures 1A–C). Further stratification of
staining pattern among cases with postive nuclear NFκB staining showed
no significant difference in clinical outcome.”
4) line 350 - the authors suggest that overexpression of NFkB might be due to
p16 inactivation - can the authors present any data to support this? eg. serial
sections stained for p65 and p16
Response:
We appreciate the reviewer for reminding us of this important issue. We made
this hypothesis that overexpression of NFκB might be due to p16 inactivation in
light of contrary correlations with survival outcomes of these two proteins by
IHC analysis. Besides, a previous research data demonstrated increased
nuclear p16 leads to ubiquitination of NFκB (reference no. 52). However, this
hypothesis was not in vitro tested in our study. For this limitation, we delete the
last sentence of the 5th paragraph of the Discussion section “The potential
interaction between p16 and NFκB is consistent with the current data and
suggest that overexpression of NFκB might come from p16 inactivation
through decreased degradation in patients treated by trimodality
treatment.”, and the second sentence of the last paragraph of the Discussion
section “Inactivation of p16 might be associated with activation of NFκB
and result in bladder tumor progression and distant metastasis.” We add
this limitation with the sentences into the 5th paragraph of the Discussion
section (Page 14, Lines 345-348). “Whether p16 inactivation plays a role in
overexpression of NFκB remains unsettled. Further in vitro analysis, such
as using p16 knock-down cell lines, to investigate the impact of p16
activation on NFκB signaling pathways to disclose the interaction is
demanded.”
5) How does the NFkB inhibitor work? This should be made clear in the text -
is this supported by the experimental data?
Response:
We appreciate the reviewer’s suggestion. The underlying molecular
mechanism of SC75741, the NFκB inhibitor used in our study, involves the
impaired DNA binding of the NFκB subunit p65. Previous study demonstrated
the inhibition of transcription activation of NFκB-dependent genes in both in
vitro and in vivo models of lung cancer, resulting in reduced expressions of
cytokines, chemokines, pro-apoptotic factors, and subsequent inhibition of
caspase activation (Cell Microbiol. 2013 Jul;15(7):1198-211). We modify the
sentences in our Methods and Materials section (Page 3, Lines 133-134).
“SC75741, an NFκB inhibitor involving impaired DNA binding of the NFκB
subunit p65 [38], was purchased from Selleck Chemicals (Houston, TX,
USA).”
6) Fig 4 - important control is missing - show that p50? is reduced in the KD
cells both by immunoblotting and by IF - why are these referred to as NFkB KD
cells? What are the consequences of p50 KD on NFKB signalling? p65 total
levels, p-p65 and localisation and transactivation of NFkB gene targets?
Response:
We thank the reviewer for reminding us of this important issue. We evaluated
the in vitro delivery efficiency of RNA interference with Western blot analysis,
which is one of the most well-established techniques for qualitatively and
quantitatively assessing gene silencing efficiency and selectivity. We added the
images of Western blot, showing the decreased p50 and p-p65 proteins of the
knock-down cells as below and in the supplementary sections (Figure S1)
NFκB is a genetic term for a family consisting of five proteins: NFκB 1
(p105/050), NFκB 2 (p100/p52), RelA (p65), RelB, and c-Rel. NFκB proteins
are usually held inactive in the cytoplasm of resting cells by association with
inhibitor of NFκB (IκB) proteins. Upon stimulation, IκB is phosphorylated by IκB
kinase complex (IKK) and leads to the activation of p65-p50 complexes. This
IKK-mediated IκB phosphorylation results in subsequent phosphorylation of
p65 and nuclear translocation of p65-p50 complexes binding to
enhancer/promoter regions of target genes (Nature Reviews Cancer.
2002;2(4):301-10; Cancer Immunology Research. 2014;2(9):823-30; Cells.
2016;5(1)). Therefore, p50 and p65 work together as a complex for downstream
NFκB signaling. If p50 could not work properly, p65-p50 complex is not formed,
and NFκB signaling would be blocked.
7) NFkB signalling has many downstream consequences/targets that could be
associated with worse prognosisis and reported in vitro observations. This
reviewer is confused by the focus on IL1B and the TME and claims made in
lines 280-282. Other downstream targets of NFKB should also be analysed.
Response:
We totally agree with the reviewer that NFκB signaling has numerous
downstream targets that correlate with cancer promotion, such as antiapoptotic factors, cell-cycle regulators, adhesion molecules, pro-inflammatory
gene induction, and various cytokines/chemokines. We also believe that a
combinatory impact of multiple events, rather than a single downstream
consequence of NFκB signaling pathway, results in bladder tumor progression
as demonstrated in our IHC analysis. In consideration of the complex
downstream signaling, the toxicities of current NFκB inhibitors for clinical
practice, and recent progress of immunotherapy in bladder cancer, we sought
to focus on the investigation of the possible mechanism regarding the
inflammatory responses and their association with tumor microenvironment.
Besides, the growing evidence suggests the interactive roles of NFκB signaling
and inflammation in tumorigenesis (Nature Immunology 2011;12(8):715-23).
Given the understanding of NFκB as an inducible rather than cell-type-specific
transcription factor that responds to proinflammatory cytokines and microbial
products, NFκB has been thought of the key regulator of inflammation (The New
England Journal of Medicine 1997;336(15):1066-71).
Among the NFκB signaling-associated inflammatory responses, IL-1β is one of
the most important targeted cytokines (Nature Immunology 2017;18(8):861-9)
promoting aggressiveness and neoangiogenesis of tumor cells (Nature
Immunology 2012;13(4):343-51). Therefore, we conducted experiments to
investigate the radiation-induced NFκB activation and IL-1β as well as its impact
on bladder cancer cells as our first step to identify the potential impact of NFκB
signaling on tumor microenvironment.
We truly appreciate the reviewer’s comment and agree that other downstream
targets (such as IL-6, IL-8, TNF-α, CSCL1, and CSCL10) should also be
analyzed. However, the investigation on the underlying mechanisms might be
beyond the main scope of this translational study on IHC staining results for
appropriate patient selection, and our ongoing work is to uncover the
complicated microenvironmental downstream signaling. We add this future
work with the sentence in the 8
th paragraph of the Discussion section
“Notably, a combinatory impact of multiple events, rather than a single
consequence of NFκB signaling pathway, results in bladder tumor
progression. Besides the interactive roles of NFκB signaling and tumor
microenvironmental inflammation in tumorigenesis, more investigational
work is ongoing for the downstream consequences/targets of NFκB”
(Page 15, Lines 387-390).
8) minor typological errors/proof reading required eg. cells were not cultured
in 95% oxygen..!
Response:
We appreciate the reviewer for correcting this error. We made this correction
in the Methods and Materials section (Page 3, Line 127).
Round 2
Reviewer 2 Report
The authors have improved the quality of their data presentation in line with the suggestions made and have adequately addressed most of the points/concerns raised.
One remaining concern, is the claim of increased nuclear NFkB staining and the evidence presented to support this. The authors have kindly provided enlarged images for Fig. 2 as requested. Nevertheless from the revised images presented in Fig. 2 it still requires considerable 'leap of faith' on the part of the reader that there is increased nuclear NFkB staining rather than increased cellular staining. Whilst, it may be logical to presume that nuclear levels may be increased (and therefore increased NFkB activity and signalling) the IHC data presented does not demonstrate this. Are cytosolic levels not increased? This is particularly important given the regulation of NFkB signalling via its intracellular localisation and also given the revised text (lines 209-211) and line 243. The scale bar for Fig 2 should also be enlarged and shown on all images. Pre-treatment and recurrent images also look as if the magnification may be different - scale bars would help here.
Importantly, immunoblots showing KD of p50 are now included.
Fig 4B images remain of poor resolution and it is unclear how cells could be accurately counted from this.
Author Response
We deeply appreciate your kind advice and informative suggestion. In response to the comments from the reviewer, we have made some revisions and resubmit a revised version according to the editorial decision. The comments and our responses are listed as follows.
Reviewer #2
One remaining concern, is the claim of increased nuclear NFkB staining and the evidence presented to support this. The authors have kindly provided enlarged images for Fig. 2 as requested. Nevertheless from the revised images presented in Fig. 2 it still requires considerable 'leap of faith' on the part of the reader that there is increased nuclear NFkB staining rather than increased cellular staining. Whilst, it may be logical to presume that nuclear levels may be increased (and therefore increased NFkB activity and signalling) the IHC data presented does not demonstrate this. Are cytosolic levels not increased? This is particularly important given the regulation of NFkB signalling via its intracellular localisation and also given the revised text (lines 209-211) and line 243.
Response:
We appreciate the reviewer’s comment. We totally agree with the reviewer’s concern that the increased cytoplasmic NFkB, rather than nuclear NFkB, was shown in our selected recurrent bladder tumor specimens. We found the contradictory findings in different studies on the localization sites of NFkB expression for the association with clinical outcomes.
The biological significance and percentage of different cellular sub-localization of NFkB p65 in carcinoma tissues, including bladder tumors, remain unsettled. A previous study showed that in muscle-invasive bladder urothelial carcinoma, nuclear overexpression of p65 was associated with poor survival outcomes, while cytoplasmic p65 did not seem to be a significant prognostic factor (Virchows Arch 2008;452(3):295-304). However, 65% of the cases in this study displayed concurrent cytoplasmic and nuclear immunoreactivity. Therefore, a plausible explanation for this finding is that both nuclear and cytoplasmic p65 might share the significance of correlation with survival outcomes.
A study on the prognostic value of different expression sites of NFkB p65 in gastric carcinoma (GC) showed that nuclear expression of p65 was significantly higher in 72% GC tissue specimens versus 17% for cytoplasmic staining (Clin Transl Sci 2013;6(5):381-385). However, the same study demonstrated contrary prognostic value of nuclear staining and cytoplasmic staining of p65. This study suggested that different cellular localizations of NFkB p65 might indicate different clinical outcomes. This was in line with our IHC results that increased nuclear p65 was correlated with worse outcomes, while cytoplasmic p65 showed no significant association with clinical outcomes. Therefore, future investigations on serial IHC stains in human bladder specimens (pre-RT, during RT, post RT, recurrent specimens if available) or mouse xenograft models might help clarify the findings.
Another study showed that in triple-negative breast cancer, 81% of the cases with overexpression of cytoplasmic p65 was associated with nuclear positivity (Br J Cancer 2007;97(4):523–530). The possible explanation for the frequent and regular accompany with enhanced nuclear staining is the limitations of the immunohistochemical method to detect nuclear translocation. This study emphasized that cytoplasmic p65 expression might be an even better ‘indirect' marker for nuclear p65 quantification than nuclear positivity, which is sometimes difficult to determine.
On the other hand, NFkB signaling might be affected in the post-irradiation microenvironment. For instance, proteasome system has been involved in the damage response and defective immune response after radiation exposure (Mol Biochem Parasitol 2017;212:55-67; Cell Res 2016;26(4):457–483). If the proteasome-ubiquitination system in bladder tumor is disturbed by radiation, p65 might accumulate in cytoplasm due to impaired degradation and/or nuclear translocation and therefore enhanced immunoreactivity in cytoplasm (Clin Cancer Res 2016;22(17):4302–4308). Besides, more radio-resistant tumor cells might be selected after irradiation, such as the cells with acetylation of p65 and the promoted transcriptional activity.
In conclusion, whether cytoplasmic expression of p65 was associated with higher rates of its presence in recurrent than primary bladder tumor samples is unclear in consideration of limited number of recurrent tumor samples in our current study. Future investigations may be needed to disclose the different relationship between primary and recurrent bladder cancers in NFkB localization and the pathophysiological significance. For this unsettled issue, we revise the phrase “Increased nuclear staining of NFkB” with “Increased NFkB staining” for the heading of Figure 2(D) legend.
The scale bar for Fig 2 should also be enlarged and shown on all images. Pre-treatment and recurrent images also look as if the magnification may be different - scale bars would help here.
Response:
We appreciate the reviewer’s suggestion. We revise the Figure 2D as below. The figure legend of figure 2D is revised as “Immunohistochemical staining of NFκB and p16 in pre-treatment and recurrent bladder tumors at magnifications of 200× (scale bar: 100 μm).
Fig 4B images remain of poor resolution and it is unclear how cells could be accurately counted from this.
Response:
We appreciate the reviewer’s suggestion. We provide the original images as below. We also revise the Figure 4B as below for better demonstration of the invading cells.
Ctrl_sham
Ctrl_RT (5Gy)
NFκB KD_sham
NFκB KD_RT (5Gy)
Gy)
This manuscript is a resubmission of an earlier submission. The following is a list of the peer review reports and author responses from that submission.
Round 1
Reviewer 1 Report
The manuscript presented by Chiang et al. aimed to investigate prognostic targets for muscle‐invasive bladder cancer. The authors observe a correlation between NF-kappaB overexpression in bladder cancer specimen and lower rates of local progression‐free survival, distant metastasis‐free survival and overall survival in patients. They further demonstrate that irradiation induced up‐regulation of NFκB signaling in vitro, which was associated with increased proliferative capability and could be reversed by applying an NFκB inhibitor or RNAi.
Although the main aim of the study seems promising, the findings presented by the author lack novelty and molecular insights justifying further consideration for publication as elucidated below.
Despite the ongoing discussion concerning the multiples roles of NFkappaB in bladder cancer, its direct implication in progression, migration and recurrence of bladder cancer is commonly accepted in the field (Carcinogenesis. 2015 Mar;36(3):299-306.) For instance, high rates of phospho- NFkappaB/p65 were already found in muscle-invasive urothelial tumors, which were directly correlated with significantly higher risks of disease progression and cancer-specific mortality (Mol Cancer Ther; 1–12. _2018, DOI: 10.1158/1535-7163.MCT-17-0786). Although the observation that NF-kappaB overexpression is directly associated with local recurrence and/or distant metastasis is still interesting, this is shown only in 4 out of 16 patients (lines 223/224) and therefore lacks evidence. Accordingly, the title of the paper is misleading, since the authors focus on the already described overexpression of NF-kappaB in bladder cancer in the patients rather than on the promotion of tumor recurrence.
In the in vitro section of the manuscript, the authors state that “NFκB signaling was upregulated after irradiation and inhibited with the treatment of an NFκB inhibitor, SC75741, mitigating the proliferative capability of bladder cancer cells”( lines 251-252). Likewise to the observations made in patients, these findings lack novelty. In particular, a range of studies already described, that NF-κB suppresses apoptosis and promotes bladder cancer cell proliferation as well as migration and metastasis in vitro and in vivo (Scientific Reports volume 7, Article number: 40723 (2017), Neoplasia. 2017 Sep; 19(9): 672–683., Cancer Letters Volume 212, Issue 1, 20 August 2004, Pages 61-70, Clinical Cancer Research Vol. 9, 2786 –2797, July 2003). Downregulation of NF-kB by respective inhibitors, as presented by the authors in the present manuscript, was further already shown to have beneficial effects for treating bladder cancer cells and reduce the tumor burden in vivo (Cancer Res. 2018 Sep 1;78(17):5182), while radiation is known to result in activation of NF-kappaB (Cancer Res. 2005;65:6601–6611.). Potential interesting mechanistic insights, for instance regarding the mechanism underlying NF-kappaB-driven tumor recurrence or potential mechanisms induced by radiation facilitating NF-kappaB overexpression are not addressed in the present manuscript. Although the potential role of p16 in bladder cancer observed here seems interesting, the manuscript likewise lacks further mechanistic insights in this direction.
In summary, although the main purpose of the study seems promising, the manuscript lacks substantial new findings or molecular insights justifying further consideration for publication.
Author Response
We very much appreciate your valuable advice and informative suggestions. We would like to respond to the critical comments by the reviewer on the novelty of our study in NFκB signaling for bladder cancer. The reviewer kindly provided a few articles with their settled knowledge of NFκB signaling in bladder cancer. Even with the published data on NFκB signaling for bladder cancer, almost all of them did not use radiation or not from patients treated with radiotherapy. Notably, our study was specific with muscle-invasive bladder cancer patients undergoing bladder-preserving radiotherapy and chemotherapy, and worked on the radiation-activated NFκB signaling in lethality of bladder cancer cells. Our study may be novel in radiation-related mechanism on NFκB signaling, and would be helpful in patient selection for bladder preservation. We truly hope for the re-consideration of the reviewer in our translational findings of radiation effect on the additional activation of NFκB signaling and the associated impact on bladder cancer cells. In response to the comments from the reviewer, we have made some revisions and resubmit a revised version according to the editorial decision. The comments and our responses are listed as follows.
Reviewer #1
The manuscript presented by Chiang et al. aimed to investigate prognostic targets for muscle‐invasive bladder cancer. The authors observe a correlation between NF-kappaB overexpression in bladder cancer specimen and lower rates of local progression‐free survival, distant metastasis‐free survival and overall survival in patients. They further demonstrate that irradiation induced up‐regulation of NFκB signaling in vitro, which was associated with increased proliferative capability and could be reversed by applying an NFκB inhibitor or RNAi.
Although the main aim of the study seems promising, the findings presented by the author lack novelty and molecular insights justifying further consideration for publication as elucidated below.
Despite the ongoing discussion concerning the multiples roles of NFkappaB in bladder cancer, its direct implication in progression, migration and recurrence of bladder cancer is commonly accepted in the field (Carcinogenesis. 2015 Mar;36(3):299-306.) For instance, high rates of phospho- NFkappaB/p65 were already found in muscle-invasive urothelial tumors, which were directly correlated with significantly higher risks of disease progression and cancer-specific mortality (Mol Cancer Ther; 1–12. _2018, DOI: 10.1158/1535-7163.MCT-17-0786). Although the observation that NF-kappaB overexpression is directly associated with local recurrence and/or distant metastasis is still interesting, this is shown only in 4 out of 16 patients (lines 223/224) and therefore lacks evidence. Accordingly, the title of the paper is misleading, since the authors focus on the already described overexpression of NF-kappaB in bladder cancer in the patients rather than on the promotion of tumor recurrence.
Response:
We appreciate these precious comments on our study from the reviewer, and fully agree with the reviewer in some researches on the association between NFκB signaling and the outcomes of bladder or other cancers.
However, our study demonstrated the first time that NFκB overexpression is associated with survival outcome in patients with muscle-invasive bladder cancer undergoing uniform bladder-preservation therapy, which consisted of maximal TURBT, chemotherapy, and radiotherapy. Given the current prognostic factors or so called selection criteria for patients with bladder-preserving therapy with exclusively clinical factors (T2-T3, absence of carcinoma in situ, hydronephrosis, or lymphadenopathy), the tumor-associated molecular markers might add prognostic implication in the successful organ preservation. Although this subgroup contains only a small proportion of muscle-invasive bladder cancer, it is important to develop potentially precise selection criteria for organ-preservation purpose and the improved quality of life. Therefore, even with broadly explored NFκB signaling in bladder cancer, our study provides a novel insight with the focus on this specific subgroup of patients. We add the proposed molecular marker for selecting specific patients undergoing bladder-preserving therapy into second paragraph of the Introduction section to expand the scope of this study (line 61-64, page 2).
With available tissues of recurrent tumors from only 4 patients, we found 3 of them with increased expression of NFκB. We understand the limitation from small number of patients with available recurrent tumor tissue to propose the association, but would like to demonstrate the post-treatment increased NFκB of these recurrent tissues for the potential implication. Pre-treatment but not post-treatment IHC staining for NFκB was the significant factor associated with the survival outcomes in our data analysis. Therefore, we did not expand the interpretation from the limited number of patients with available recurrent tumor tissues. To propose the potential link, we supplement the in vitro data on irradiated bladder cancer cells with increased NFκB expression and enhanced invasiveness and clonogenic survival. These findings might still imply the importance of radiation activated NFκB signaling for the resistance of bladder cancer, and support the rationale of future intervention of NFκB for bladder preservation with radiotherapy and other treatments. We add one paragraph of the limited conservative interpretation from NFκB of recurrent tumor tissues into the Discussion section (line 362-372, page 14).
As the reviewer kindly suggested for the previous publications on NFκB, we look through these studies. Very few of them actually worked on NFκB pathway activated by radiation in bladder cancer. We would like to list them and compare their details with our study as follows.
1. Nuclear factor-kB promotes urothelial tumorigenesis and cancer progression via cooperation with androgen receptor signaling. Mol Cancer Ther. 2018 Jun;17(6):1303-1314
Satoshi Inoue et al. demonstrated that patients with NF-κB/p65 positivity by immunohistochemistry had significant higher risks of disease progression.
All patients in this study underwent cystectomy. Instead, our study focused on the patients undergoing radiation-based treatment, which is a subgroup of patients with a different treatment modality from surgical cystectomy alone.
2. To be an ally or an adversary in bladder cancer: the NF-κB story has not unfolded; Carcinogenesis. 2015 Mar;36(3):299-306
This is a review article on the correlation between NF-κB signaling and bladder cancer.
None of the studies in this review article included patient groups undergoing bladder-preserving therapy, and none of them investigated the direct association of radiation with NF-κB pathway in bladder cancer.
In the in vitro section of the manuscript, the authors state that “NFκB signaling was upregulated after irradiation and inhibited with the treatment of an NFκB inhibitor, SC75741, mitigating the proliferative capability of bladder cancer cells”( lines 251-252). Likewise to the observations made in patients, these findings lack novelty. In particular, a range of studies already described, that NF-κB suppresses apoptosis and promotes bladder cancer cell proliferation as well as migration and metastasis in vitro and in vivo (Scientific Reports volume 7, Article number: 40723 (2017), Neoplasia. 2017 Sep; 19(9): 672–683., Cancer Letters Volume 212, Issue 1, 20 August 2004, Pages 61-70, Clinical Cancer Research Vol. 9, 2786 –2797, July 2003). Downregulation of NF-kB by respective inhibitors, as presented by the authors in the present manuscript, was further already shown to have beneficial effects for treating bladder cancer cells and reduce the tumor burden in vivo (Cancer Res. 2018 Sep 1;78(17):5182), while radiation is known to result in activation of NF-kappaB (Cancer Res. 2005;65:6601–6611.). Potential interesting mechanistic insights, for instance regarding the mechanism underlying NF-kappaB-driven tumor recurrence or potential mechanisms induced by radiation facilitating NF-kappaB overexpression are not addressed in the present manuscript. Although the potential role of p16 in bladder cancer observed here seems interesting, the manuscript likewise lacks further mechanistic insights in this direction.
In summary, although the main purpose of the study seems promising, the manuscript lacks substantial new findings or molecular insights justifying further consideration for publication.
Response:
We totally respect the reviewer’s criticism and agree that there was a lot of room for improvement of our in vitro investigation. Our study is the first study, as we can approach, to investigate the potential molecular markers in patients undergoing bladder-preservation therapy by IHC and in vitro study. Therefore, the contribution of our patient and in vitro study was an attempt to demonstrate the evident radiation-induced NFκB signaling and aggressiveness of bladder cancer. However, we fully agree with the reviewer’s suggestion in the unmet need for the true mechanism of radiation associated mechanism of NFκB activation. This study started with the discovery part in molecular markers from tissue IHC for their association with survival outcomes of patients with muscle-invasive bladder cancer undergoing uniform bladder-preservation therapy. The translational part following the IHC findings with NFκB confirmed the radiation activated NFκB signaling associated with resistance of bladder cancer cells. The underlying mechanism on additional radiation-activated NFκB to the pre-existing overexpressed NFκB signaling remains to be explored. We add this limitation and future perspective to the limitation paragraph of Discussion section (line 356-361, page 14).
Although plenty of previous studies explored the correlation between NFκB signaling and bladder cancer, as the reviewer kindly provided, most of the studies did not include the role of irradiation. Besides, many other researches focused on the link between irradiation and NFκB signaling but used other cancers than bladder cancer cell lines or manipulated/selected bladder cancer cell lines, such as cisplatin-resistant bladder cancer cell line or BBN-induced highly aggressive bladder cancer cell line. Therefore, previous studies were not capable of reflecting the direct effect of irradiation-triggered NFκB signaling on bladder cancer.
We read over the studies the reviewer kindly provided as below, but found none of them connecting NFκB pathway with radiation in bladder cancers.
3. NF-κB suppresses apoptosis and promotes bladder cancer cell proliferation by upregulating survivin expression in vitro and in vivo; Sci Rep. 2017 Jan 31;7:40723;
Xiaolu Cui et al demonstrated the carcinogenic function of the NF-κB/survivin pathway in bladder cancer using a xenograft tumor model of stable NF-κB overexpression bladder cancer cells.
No radiation was involved in this study
4. NF-κB p65 overexpression promotes bladder cancer cell migration via FBW7-mediated degradation of RhoGDIα protein; Neoplasia. 2017 Sep;19(9):672-683;
Junlan Zhu et al. demonstrated that p65 overexpression resulted in promotion of bladder cancer using BBN-induced high invasive bladder cancer cell lines.
No radiation was involved in this study
5. Up-regulation of Bfl-1/A1 via NF-kB activation in cisplatin-resistant human bladder cancer cell line; Cancer Lett. 2004 Aug 20;212(1):61-70;
Jin Koo Kim et al. showed that NF-kB inhibited cisplatin- and TNF-alpha-induced apoptosis by upregulation of Bfl-1/A1 pathway using cisplatin-resistant bladder cancer cell line
No investigation between radiation and NF-κB signaling was discussed in this study
6. Nuclear factor-κB mediates angiogenesis and metastasis of human bladder cancer through the regulation of interleukin; Clin Cancer Res. 2003 Jul;9(7):2786-97
Takashi Karashima et al. showed that acidic and hypoxic condition increased IL-8 using human TCC cell lines.
No radiation was involved in this study
7. Loss of tumor suppressor p53 decreases PTEN expression and enhances signaling pathways leading to activation of Activator Protein 1 and nuclear factor-κB Induced by UV radiation. Cancer Res. 2005 Aug 1;65(15):6601-11
Jian Wang et al showed that p53 had a suppressive activity on the cell signaling pathways leading to decrease in UV-induced activation of NF-κB using mouse epidermal C141 cells
No bladder cancer cells were used in this study
8. Editor's Note: Curcumin potentiates the antitumor effects of Bacillus Calmette-Guerin against bladder cancer through the downregulation of NF-kB and upregulation of TRAIL receptors. Cancer Res. 2018 Sep 1;78(17):5182
This was an Editor’s Note, rather than the original research, with the findings in this review articles not sufficient to determine the relationship between the images and the experiments described.
In this study, Ashish Kamat et al. showed that curcumin suppressed the BCG-induced activation of cell survival transcription factor using syngeneic bladder cancer model.
No investigation between radiation and NF-κB signaling was discussed in this study.
Reviewer 2 Report
Congratulations for the well written paper and interesting results.
Minor comments:
Overexpression of “NFκB” should be specified in the abstract (staining for p65)?
Obviously whole mount sections were immunostained (no TMA): can the staining pattern be further specified, i.e. uniform or inhomogenous staining for NFκB?
P2 L95: what kind of autostainer?
Polyclonal antibody NFκB p65 (A) used in the study has been discontinued by the manufacturer and replaced by monoclonal NFκB p65 (F-6).
NFκB KD cell line targeted at p50, antibody is p65, please discuss.
Table 3 inconvenient to read in present form EGFR multivariate analysis missing.
Uniform use of „p16“ with lower case “p”.
PD-L1 CPS score missing.
Tendency p53 for LPFS, DMFS and OS (in multivariate analysis for LPFS even 0.05): small number of cases. Has
In vitro assay suggests role of p-p65, cases p65. Please discuss.
P14 L307-309 “We may deduce from these researches and our study that radiation may cause suppressed apoptosis by activation of NFκB signaling, and results in somatic resolution and more lethal manner in bladder cancer. “ This sentence is difficult to understand and needs to be reworded.
Minor spelling errors, see attached file with corrections.
Author Response
We deeply appreciate your kind advice and informative suggestion. In response to the comments from the reviewer, we have made some revisions and resubmit a revised version according to the editorial decision. The comments and our responses are listed as follows.
Reviewer #2
Congratulations for the well written paper and interesting results.
Minor comments:
Overexpression of “NFκB” should be specified in the abstract (staining for p65)?
Response:
We thank the reviewer for the kind advice and suggestion. We add the information of p65 staining into line 27-40, page 1.
Obviously whole mount sections were immunostained (no TMA): can the staining pattern be further specified, i.e. uniform or inhomogenous staining for NFκB?
Response:
We appreciate the reviewer’s suggestion for specifying the staining pattern. Among the cases with positive NFκB staining, the stratification by staining pattern (score 1: scattered single cells; score 2: patchy with sharp demarcation; score 3: patchy with gradual change; score 4: diffuse and homogenous) did not show significant difference in clinical outcome. We add this information into line 108-110, page 3 and line 205-212, page 6.
P2 L95: what kind of autostainer?
Response:
We thank the reviewer for reminding us of this important issue. The autostainer used in this study was Ventana Benchmark XT (Ventana Medical Systems, Tucson, AZ, USA). We add this information into line 95-96, page 3.
Polyclonal antibody NFκB p65 (A) used in the study has been discontinued by the manufacturer and replaced by monoclonal NFκB p65 (F-6).
Response:
We appreciate the reviewer for pointing out this important information. After checking with our pathologist, we confirm that the antibody of NFκB p65 used in this study was monoclonal NFκB p65 (F-6) (Santa Cruz Biotechnology, Santa Cruz, CA, USA) as the reviewer specified. We made the correction in our revised Supplementary Table 1.
NFκB KD cell line targeted at p50, antibody is p65, please discuss.
Response:
NFκB is a genetic term for a family consisting of five proteins: NFκB 1 (p105/050), NFκB 2 (p100/p52), RelA (p65), RelB, and c-Rel. NFκB proteins are usually held inactive in the cytoplasm of resting cells by association with inhibitor of NFκB (IκB) proteins. Upon stimulation, IκB is phosphorylated by IκB kinase complex (IKK) and leads to the activation of p65-p50 complexes. This IKK-mediated IκB phosphorylation results in subsequent phosphorylation of p65 and nuclear translocation of p65-p50 complexes binding to enhancer/promoter regions of target genes (Nature reviews Cancer. 2002;2(4):301-10; Cancer immunology research. 2014;2(9):823-30; Cells. 2016;5(1)). Therefore, p50 and p65 work together as a complex for downstream NFκB signaling. If p50 could not work properly, p65-p50 complex is not formed, and NFκB signaling would be blocked. The standard protocol for NFκB immunohistochemistry in our department of pathology contains the antibody p65, which we used in our study. In our in vitro study, we tried to use two NFκB knock down (KD) cell lines targeted at p65 and p50, respectively. However, the effect of RNA interference of NFκB KD cell line targeted at p65 was not stable in the designed conditions, and we are still working on the technical reasons. In the meantime, we investigate our IHC findings by using NFκB cell line targeted at p50.
Table 3 inconvenient to read in present form EGFR multivariate analysis missing.
Response:
We thank the reviewer for the comment. We have revised the format of Table 3 as below and in the main text.
Uniform use of „p16“ with lower case “p”.
Response:
We appreciate this reviewer’s comment. We have uniformed the use of p16 in Table 2, Table 3, and Supplementary Table 2 of the revised manuscript.
PD-L1 CPS score missing.
Response:
We thank the reviewer for the kind advice. However, unlike clone 22C3 or 28-8, the percentages of TC and IC were scored separately for clone SP142, which was the clone of PD-L1 antibody used in this study. Therefore, the combined positive score (CPS) is not applicable to this clone according to the manufacturer's instruction.
Tendency p53 for LPFS, DMFS and OS (in multivariate analysis for LPFS even 0.05): small number of cases. Has
Response:
We fully agree with the reviewer on the connection between p53 staining and clinical outcome in this study, in terms of the trend of p53 with LPFS, DMFS, and OS in univariate analysis and the significance with LPFS in multivariate analysis. Besides, previous studies demonstrated the association of p53 with the outcomes of bladder cancer patients undergoing radical cystectomy (J Clin Oncol. 2004 Mar 15;22(6):1014-24) and of patients with non-metastatic advanced urothelial bladder cancer (BJU Int. 2010 Feb;105(4):489-95). However, we failed to show such the potential association between p53 and clinical outcomes even with the changed cut-off point of p53 positivity or with the integration of p53 into a panel of biomarkers. We add this information as below and into line 205-212, page 6.The possible reasons for no association may include small sample size or limited information of section tissues from TURBT.
|
molecular panel |
LPFS |
DMFS |
OS |
||||
|
Univariate |
univariate |
univariate |
|||||
|
3y/5y (%) |
P value |
3y/5y (%) |
P value |
3y/5y (%) |
P value |
||
|
p53/p16 |
p53 negative+ p16 negative others |
73/65 80/67 |
0.58 |
86/86 80/76 |
0.22 |
93/86 79/75 |
0.20 |
|
p53 positive+ p16 positive others |
50/50 78/69 |
0.26 |
50/50 81/75 |
0.76 |
50/50 89/80 |
0.58 |
|
|
p53/ NFκB |
p53 negative+ NFκB positive others |
81/73 75/64 |
0.66 |
82/82 83/80 |
0.44 |
91/82 96/78 |
0.36 |
|
p53 positive+ NFκB negative others |
66/44 76/60 |
0.58 |
75/75 85/82 |
0.53 |
66/66 88/81 |
0.52 |
|
|
p53/p16/ NFκB |
p53/p16 negative+ NFκB positive others |
60/40 80/70 |
0.06 |
60/60 85/82 |
0.51 |
80/60 88/82 |
0.52 |
|
p53/p16 positive+ NFκB negative others |
95/95 73/67 |
0.75 |
95/95 81/78 |
0.43 |
95/95 84/78 |
0.47 |
|
In vitro assay suggests role of p-p65, cases p65. Please discuss.
Response:
In the resting cells, NFκB proteins (p50 and p65) remain inactive in the cytoplasm by association with IκB. Upon stimulation, IκB is phosphorylated by IκB kinase complex (IKK) and leads to the activation of p65-p50 complexes. This IKK-mediated IκB phosphorylation results in subsequent phosphorylation of p65 (p-p65), followed by nuclear translocation of p65-p50 complexes binding to enhancer/promoter regions of target genes (Nature reviews Cancer. 2002;2(4):301-10; Cancer immunology research. 2014;2(9):823-30; Cells. 2016;5(1)). When analyzing p65 by IHC, we may visualize the expression of p65 both in nucleus and in cytoplasm. Besides, p-p65 is prone to further degradation, therefore, may not be a suitable target for IHC. p-p65, as compared to p65, might better reflect the true activation of NFκB signaling since the expression of p-p65 illustrates the transient period for p65-p50 complex from cytoplasm to nucleus for the effect of radiation on bladder cancer cell lines by western blot or immunofluorescence.
P14 L307-309 “We may deduce from these researches and our study that radiation may cause suppressed apoptosis by activation of NFκB signaling, and results in somatic resolution and more lethal manner in bladder cancer. “ This sentence is difficult to understand and needs to be reworded.
Response:
Thank the reviewer for the suggestion. We re-wrote the sentence (line 316-317, page 13) as “We may deduce from these researches and our study that radiation may induce activated NFκB signaling, which prevents subsequent lethal cascade in bladder cancer.”
Round 2
Reviewer 1 Report
However, our study demonstrated the first time that NFκB overexpression is associated with survival outcome in patients with muscle-invasive bladder cancer undergoing uniform bladder-preservation therapy, which consisted of maximal TURBT, chemotherapy, and radiotherapy. Given the current prognostic factors or so called selection criteria for patients with bladder-preserving therapy with exclusively clinical factors (T2-T3, absence of carcinoma in situ, hydronephrosis, or lymphadenopathy), the tumor-associated molecular markers might add prognostic implication in the successful organ preservation. Although this subgroup contains only a small proportion of muscle-invasive bladder cancer, it is important to develop potentially precise selection criteria for organ-preservation purpose and the improved quality of life. Therefore, even with broadly explored NFκB signaling in bladder cancer, our study provides a novel insight with the focus on this specific subgroup of patients. We add the proposed molecular marker for selecting specific patients undergoing bladder-preserving therapy into second paragraph of the Introduction section to expand the scope of this study (line 61-64, page 2).
With available tissues of recurrent tumors from only 4 patients, we found 3 of them with increased expression of NFκB. We understand the limitation from small number of patients with available recurrent tumor tissue to propose the association, but would like to demonstrate the post-treatment increased NFκB of these recurrent tissues for the potential implication. Pre-treatment but not post-treatment IHC staining for NFκB was the significant factor associated with the survival outcomes in our data analysis. Therefore, we did not expand the interpretation from the limited number of patients with available recurrent tumor tissues. To propose the potential link, we supplement the in vitro data on irradiated bladder cancer cells with increased NFκB expression and enhanced invasiveness and clonogenic survival. These findings might still imply the importance of radiation activated NFκB signaling for the resistance of bladder cancer, and support the rationale of future intervention of NFκB for bladder preservation with radiotherapy and other treatments. We add one paragraph of the limited conservative interpretation from NFκB of recurrent tumor tissues into the Discussion section (line 362-372, page 14).
The reviewer agrees, that the findings described here, although being well-known in the field for bladder cancer and other cancer specimen, may have been not observed in this specific sub-population of bladder cancer. The reviewer further agrees that developing precise selection criteria for organ-preservation purpose and improved quality of life is highly important. However, despite the point raised by the authors that pre-treatment but not post-treatment IHC staining for NFκB was the significant factor associated with the survival outcomes, the findings are still limited to a very small number of patients. This small number of patients with recurrent tumors and increased overexpression of NF-kB after bladder-preservation therapy (3 out of 4 in total) is still too low to support the major hypothesis that NF-kappaB overexpression is 1.) driven by bladder-preservation therapy and 2.) directly associated with local recurrence and/or distant metastasis in a irradiation-dependent manner. Accordingly, the title of the paper is still not suffficiently supported by the observations and thus misleading and should be definitely revised.
As the reviewer kindly suggested for the previous publications on NFκB, we look through these studies. Very few of them actually worked on NFκB pathway activated by radiation in bladder cancer. We would like to list them and compare their details with our study as follows.
We totally respect the reviewer’s criticism and agree that there was a lot of room for improvement of our in vitro investigation. Our study is the first study, as we can approach, to investigate the potential molecular markers in patients undergoing bladder-preservation therapy by IHC and in vitro study. Therefore, the contribution of our patient and in vitro study was an attempt to demonstrate the evident radiation-induced NFκB signaling and aggressiveness of bladder cancer. However, we fully agree with the reviewer’s suggestion in the unmet need for the true mechanism of radiation associated mechanism of NFκB activation. This study started with the discovery part in molecular markers from tissue IHC for their association with survival outcomes of patients with muscle-invasive bladder cancer undergoing uniform bladder-preservation therapy. The translational part following the IHC findings with NFκB confirmed the radiation activated NFκB signaling associated with resistance of bladder cancer cells. The underlying mechanism on additional radiation-activated NFκB to the pre-existing overexpressed NFκB signaling remains to be explored. We add this limitation and future perspective to the limitation paragraph of Discussion section (line 356-361, page 14).
Although plenty of previous studies explored the correlation between NFκB signaling and bladder cancer, as the reviewer kindly provided, most of the studies did not include the role of irradiation. Besides, many other researches focused on the link between irradiation and NFκB signaling but used other cancers than bladder cancer cell lines or manipulated/selected bladder cancer cell lines, such as cisplatin-resistant bladder cancer cell line or BBN-induced highly aggressive bladder cancer cell line. Therefore, previous studies were not capable of reflecting the direct effect of irradiation-triggered NFκB signaling on bladder cancer.
We read over the studies the reviewer kindly provided as below, but found none of them connecting NFκB pathway with radiation in bladder cancers.
Although the role of NF-kB in bladder cancer patients as well as the activation of NF-kB by irradiation are well known, the reviewer agrees that only very few studies worked on the activation of NFκB by radiation in muscle-invasive bladder cancer in the patient. The authors state, that “many other researches focused on the link between irradiation and NFκB signaling but used other cancers than bladder cancer cell lines or manipulated/selected bladder cancer cell lines, such as cisplatin-resistant bladder cancer cell line or BBN-induced highly aggressive bladder cancer cell line”. However, the authors also only used an immortal cell line (Human bladder urothelial carcinoma cell line, T24) in the present study. To which extend does this cellular model reflect muscle-invasive bladder cancer in the patient? The authors should provide respective literature here or focus on a more suitable in vitro model system, e.g. using primary muscle-invasive bladder cancer cells. This issue is of particular importance, since the effects of raditation on NF-kB demonstrated in the present study were already assessed in bladder cancer cell lines, as depicted above by the authors. Alternatively, the authors may investigate underlying molecular mechanism as already asked in the first revision.